# Effect of Simulated Mastication on Structural Stability of Prosthetic Zirconia Material after Thermocycling Aging

**DOI:** 10.3390/ma16031171

**Published:** 2023-01-30

**Authors:** Anna Ziębowicz, Bettina Oßwald, Frank Kern, Willi Schwan

**Affiliations:** 1Department of Biomaterials and Medical Devices Engineering, Silesian University of Technology, 41-800 Zabrze, Poland; 2Institute for Manufacturing Technologies of Ceramic Components and Composites (IFKB), University of Stuttgart, D-70569 Stuttgart, Germany

**Keywords:** zirconia, microstructure, phase composition, chewing simulation, accelerated aging, impact protectors

## Abstract

Recent trends to improve the aesthetic properties—tooth-like color and translucency—of ceramic dental crowns have led to the development of yttria-stabilized zirconia (Y-TZP) materials with higher stabilizer content. These 5Y-TZP materials contain more cubic or t’ phase, which boosts translucency. The tradeoff as a consequence of a less transformable tetragonal phase is a significant reduction of strength and toughness compared to the standard 3Y-TZP composition. This study aims at determining the durability of such 5Y-TZP crowns under lab conditions simulating the conditions in the oral cavity during mastication and consumption of different nutrients. The test included up to 10,000 thermal cycles from 5 °C to 55 °C “from ice cream to coffee” and a chewing simulation representing 5 years of use applying typical loads. The investigation of the stress-affected zone at the surface indicates only a very moderate phase transformation from tetragonal to monoclinic after different varieties of testing cycles. The surface showed no indication of crack formation after testing. It can, therefore, be assumed that over the simulated period, dental crowns of 5Y-TZP are not prone to fatigue failure.

## 1. Introduction

The specific nature of the oral cavity environment defines many requirements that prosthetic restorations should meet. One of the methods of replacing missing teeth is a crown (also known as a cup). Crowns are used in the event of significant tooth destruction, tooth fracture, chipping, discoloration, or in the absence of a tooth crowns based on implants. Crowns can be made of various materials: full ceramic, porcelain on metal or zirconium oxide foundation, composite, and acrylic (temporary crowns) [1,2,3]. Ceramic crowns cover entire teeth to correct misshapen, crooked, and damaged teeth and provide a solution that’s indistinguishable from natural teeth. In some instances, they’re even more cost-effective than metal crowns. Ceramics tend to be resistant to wear as opposed to other materials. The representative of the group of ceramic materials is zirconium oxide. That is the material of choice, inert to the body, not causing allergic reactions, and has the greatest aesthetics. From the beginning of the digitization of dentistry, the quality criteria for the production and processing of zirconium oxide have been analyzed and discussed in numerous publications. Although it has long been the most popular material for dental prostheses, you can often get the impression that you need to be on your guard all the time, as if it were a questionable material.

Understanding and knowledge of zirconia in relation to material behavior and its treatment and processing have evolved in recent years—an evolution toward a very aesthetically demanding end customer, but also an absolutely stable and safe substitute material [4].

The excellent mechanical properties of partially stabilized zirconium dioxide ceramics are due to transformation toughening, i.e., the stress-induced transformation of metastable tetragonal phase to a stable monoclinic phase which results in volume expansion and shear, and a reduction of stress intensity at the crack tip [5,6,7,8]. Metastability of the tetragonal phase at ambient temperature is achieved by the addition of stabilizers, such as yttrium, cerium, or calcium oxide [9]. These oxides stabilize the tetragonal phase by inducing oxygen vacancies and/or expanding the zirconia lattice. The metastability of the tetragonal phase is, therefore, beneficial with regard to mechanical properties. However, especially in the case of yttria-stabilized zirconia, it is also the cause of low-temperature degradation (LTD), which is a non-stress-induced slow transformation phenomenon caused by the uptake of oxygen-containing molecules and elimination of the phase stabilizing oxygen vacancies [10]. Mechanical stress during mastication and contact with saliva may cause a cooperative disruption of the surface structure of zirconia crowns [11,12,13,14].

In the last few years, in addition to the well-tested and well-documented 3Y-ZP-type zirconias, a 3 mol% yttria (Y_2_O_3_) stabilized tetragonal zirconia polycrystals, a material with high strength and moderate toughness, many new and innovative types of blanks have appeared on the market. The new materials address issues such as translucency and more tooth-like coloring. Due to these recent developments, it is more and more difficult to trace the nature and suitability of their application. Consequently, especially for dentists and patients, it becomes increasingly complicated to understand the effects of compositional changes and to clearly define and differentiate between different material qualities and concepts.

Zirconium oxide blocks with increased translucency, optimized for monolithic restorations, have recently become more and more popular. In this context, the relationship between increasing translucency, on the one hand, and decreasing strength, on the other hand, is discussed many times in specialist articles and congresses [15,16,17,18,19,20,21,22]. Super-high translucent zirconia of the 5Y-TZP-type has been on the market since around 2014, and the use is mostly limited to 3-unit bridges due to reduced strength. The basic idea behind the 5Y-TZP is to increase the content of the cubic phase by higher stabilizer content. The effect on translucency is that the cubic phase—contrary to tetragonal—is not birefringent and that in cubic grains, there is no solute drag which allows the formation of larger grains and thereby reduces scattering at grain boundaries. The downside of the concept is that the cubic phase shows no transformation toughening effect and is inherently less damage-tolerant. This negatively affects both strength and fracture resistance.

Therefore, scientists are asking themselves about the possibility of assessing the quality of the latest materials enabling the construction of 14-point bridges (ceramics of type II class 5 in accordance with ISO 6872: 2019-01 [23]), offering at the same time: significantly increased aesthetic properties and sufficient safety.

The bending strength expressed in MPa is supposed to be an important quality criterion for the applicability of a certain material. Strength, according to Griffith, scales with fracture resistance and scales inversely with the square root of the defect size. As the probability of a critical defect in a brittle material scales with the stressed volume (Weibull theory), the three approved testing protocols in ISO 23242:2020 (three-point, four-point, and biaxial) based on different sample geometries lead to different strength values for the same material [24]. The values listed in data sheets and standards, e.g., in the case of bridges larger than four elements, the value of 800 MPa, must be considered an orientational value rather than a strict design criterion [25]. In this context, high initial values of the forces required to crush materials (measured in N) provide sufficient stability (>1000 N) even after the aging process, for example, to withstand the bruxism’s maximum force.

As Chevalier et al. has shown [26], dental implants and restorations are exposed to alternating load during mastication, and it is rather the fatigue and not the bending strength tested in a single catastrophic event that determines the durability of such components. Therefore, in order to give an authoritative forecast on the fatigue strength σ_F_, besides bending strength σ also fracture toughness (critical stress intensity factor *K_IC_* [MPa√m]) and the resistance to subcritical crack growth (*K_I_*_0_, the “threshold toughness” [MPa√m]) have to be considered. *K_IC_* is measured in a single catastrophic failure test. *K_I_*_0_ corresponds to the stress intensity level up to which a material can be continuously or alternately loaded without incurring crack growth. If *K_I_*_0_ is exceeded, the crack grows such that under fatigue conditions, the fatigue strength compared to bending strength is reduced by a factor (*K_I_*_0_/*K_IC_*).
(1)$$\sigma_{k}=\left(\frac{K_{I0}}{K_{IC}}\right)*\;\sigma \;$$

While *K_IC_* values are often listed (recommended but now mandatory), the more relevant *K_I_*_0_ value, which is much more difficult to determine (e.g., by double torsion), is not part of the datasheet. It should further be mentioned that *K_I_*_0_ is not a material property but depends strongly on the test protocol. It has been shown that for 3Y-TZP, *K_I_*_0_ = *K_IC_* = 5 MPa√m under perfectly dry conditions, while *K_I_*_0_ is reduced to ~3 MPa√m in a moist environment such as in the patient’s mouth. This reduction is due to crack tip corrosion under slow fracture conditions. Hence, according to Equation (1), this means that the fatigue strength is only ~60% of the inert strength. Considering that, in the best case, the biting force of a human may be as high as 600 MPa, 60% of 800 MPa = 480 MPa, as specified, may not be enough.

The difference *K_IC_*-*K_I_*_0_ is often called the “R-curve related toughness,” i.e., the toughness increment that is caused by toughening effects, in the case of Y-TZP, predominantly by transformational toughening. Consequently, 5Y-TZP materials with a considerably lower amount of transformable tetragonal phase do not only show lower *K_IC_* at almost unaltered *K_I_*_0_. Still, with respect to fatigue strength, as Jerman et al. [27] have demonstrated, this does not compensate for the lower inert strength.

We may summarize that prediction of the longevity of dentures based on available data is difficult. Unfortunately, not all necessary data are listed, data may be different depending on measurement protocol, and there is always some risk that measurements were not carried out correctly.

Experimental approaches to measure the durability of prosthetic devices under conditions close to reality are still justified and contribute significantly to the evaluation of such materials [28,29,30,31].

The long-term behavior of the material is analyzed in simulated chewing stress tests, which are more practical than standard material tests. Chewing simulation provides a test procedure that allows reliable conclusions to be drawn about the behavior of the material in its clinical application [32,33,34]. This simulates the typical aging process [35,36,37,38,39] of a prosthetic restoration in an artificial environment of the oral cavity. Crowns are exposed to the influence of moisture, and alternating thermal (hydrothermal) stresses under the influence of the mechanical load of chewing. Chewing simulation—simulates the clinical behavior of 1,200,000 × 50 N over thermal cycles of 5–55 °C. This is equivalent to five years in the patient’s mouth (study conducted by University Hospital Regensburg, 07/2018) [40,41].

The main purpose of the research was the assessment of the impact of the cyclic temperature change and the chewing simulation process on the stability of the yttrium-stabilized zirconia material structure: 5Y-TZP.

## 2. Materials and Methods

### 2.1. Sample Preparation in CAD/CAM

In this study, one CAD/CAM zirconia 5Y-TZP block was used (Bloomden, Irvine, CA, USA). The procedure to manufacture included the following stages: (a) a model of the anatomical crown was randomly selected to produce a 3D data set; (b) arrangement of the maximum number of crowns in the block was made (it is generally accepted that one designed crown should have three connectors, not intersecting and located in easily accessible places in order to remove them after the milling process is completed—Figure 1); hyperDENT software was used (FOLLOW-ME! Technology Group, Munich, Germany); (c) the resulting.nc file was transferred to a 5-axis magnetic milling machine RS5 (RS-Team, Otwock, Andrespol, Poland) that performed the manufacturing process [42].

### 2.2. Thermocycling Aging

The crowns were cleaned of dust remaining after milling using compressed air and sintered at 1530 °C (Robocam furnace, Warsaw, Poland). The total sintering time was 7 h 57 min. After sintering, the samples were subjected to a thermocyclic aging process. This was to achieve a realistic simulation of oral temperature changes during eating. The crowns were exposed to 5000 and 10,000 thermal cycles [43] between 5 °C and 55 °C water temperature changes with a dwell time of 20 s per bath in the thermocycler (Model 1140, SD Mechatronik GmbH, Feldkirchen-Westerham, Germany). The total process time for 10,000 cycles was 98 h.

### 2.3. Chewing Simulation

For testing the SD Mechatronik chewing simulator CS-4.4. was used (Feldkirchen-Westerham, Germany), shown in Figure 2. The chewing simulator has two moving parts, the vertical bar (*z*-axis) and the horizontal table (*x*-axis). The samples were mounted to the table, which can move back and forth. Because of the two axes, there are two kinds of possible movements. The linear and the circular movement. In this study, a linear movement (Figure 3) was used to determine the resistance of the tested crowns (evaluation of the wear behavior of crowns regarding the possibility of the bite through and fracture resistance) with the use of steel antagonist (cone 30°; tip radius 1 mm). Further research using the natural shape of the tooth as an antagonist is planned, and the clinical importance of chewing simulation will certainly be increased.

The load applied to the examined crowns was 50 N (chambers I and IV) and 70 N (chambers II and III), determined on the basis of literature data [44] and explained in Figure 3, created by weights applied to the samples by the antagonists.

In the chewing simulation, mainly two-body-wear in combination with fatigue occurs. In general, fatigue occurs when a material is subjected to repeated loading and unloading. If the loads exceed a certain threshold on stress concentrators, which include, among others, slip bands (PSBs) and grain interfaces, microscopic cracks will start to form [45]. Eventually, a crack will reach a critical size, the crack will suddenly propagate, and the structure will fracture. In the case of anatomical crowns with smooth transitions and rounded surfaces of the grooves, such shapes do not reduce their fatigue strength.

Two simulation processes were performed. The first concerned the assessment of the impact of the load on the crowns in the initial state—subjected only to the sintering process. In the second, the samples after thermal aging were tested. The assumption of the chewing simulation process was to determine the impact of loading on the samples in the initial state (sintering) and after aging (for 5000 and for 10,000 thermocycles). With reference to the literature data [41,42], the number of chewing cycles simulating 5 years of use of the prosthetic device was 1,200,000. One of the crowns (placed in chamber IV/B4_in_chsx2_50N) was subjected to a double process (with the assumption of long-term use—10 years).

The legend of the designations of the samples subjected to the simulation is presented in Table 1.

### 2.4. Microscopy Observation

Observations of the surfaces of the entire crowns subjected to the chewing simulation process were carried out using stereo zoom microscopy (Nikon SMZ-10A, Tokyo, Japan) with the use of a total magnification range from 7.5× to 49×.

Using a scanning electron microscope (Jeol JSM-7001F, Boston, MA, USA), the wear morphology of zirconia ceramic crowns after contact with the opposing “teeth” was also assessed. The tested samples were not covered with a layer of conductive material. Imaging was performed under reduced vacuum using a vCD (BSE—backscattered electron imaging) detector.

The scanning microscope was also used to evaluate the microstructure of the tested materials. In this case, the surfaces of the crowns were covered with a Pt-Pd (80:20) conductive layer with a thickness of 10 nm.

### 2.5. XRD

The XRD examination was carried out on the non-etched surfaces of the cut-off part of the crowns released from the resin after the completion of the aging and chewing simulation process. Surfaces showing satisfactory smoothness were selected for the tests (Figure 4). In order to determine the impact of the applied processes, since the essence of low-temperature processes on the structural stability of the tested material, the transformation of the metastable tetragonal phase into the monoclinic phase of zirconium oxide was identified by testing the phase composition of the near-surface layer of the material samples. The crystalline phases of the sinterized crowns, after 5000 and 10,000 cycles of thermocycling aging surfaces and after chewing simulation, were characterized by X-ray diffraction analysis (XRD) using an X’Pert MPD (Malvern Panalytical, Royyston, UK, Bragg Brentano setup, CuKα1) operated at 40 kV and 40 mA, with a scan rate of 0.2°/min.

For the determination of the monoclinic content, the three characteristic peaks in the 26–33° 2θ-range were integrated, and the monoclinic content was determined according to the calibration curve of Toraya [46]. Transformation zone sizes were calculated according to Kosmac [47]. As the (101) tetragonal and (111) cubic peaks at ~30° 2θ coincide and cannot be separated, the cubic and tetragonal contents were analyzed by integrating the tetragonal (004) and (400) peaks and the cubic (400) peak in the 72–76° 2θ-range.

### 2.6. Thermal Etching

Thermal etching technology is being widely used in the analysis of ceramic microstructures and has become an effective method of preparing SEM samples. After XRD analysis had been completed to reveal the microstructure of the material, samples were subsequently lapped and polished with a 15 µm, 6 µm, and 1 µm diamond. After that, a non-crystalizing silica polishing suspension was used until a mirror-like surface was achieved. Polished samples were etched in hydrogen at the thermal etching temperature T_te_ = 1300 °C for 10 min to reveal the grain boundaries (Xerion, Freiberg, Germany). That temperature was about 200 °C lower than the ceramic sinterization temperature (T_s_ = 1500 °C).

The microstructure was studied by SEM (Jeol JSM-7001F, Boston, MA, USA) in lens SE mode, 10 kV acceleration voltage).

## 3. Results

### 3.1. Microscopy Observation

The images obtained as a result of the observations are summarized in Table 2. The rusty color of the abrasion is caused by the dissolution of the metal material in an environment of artificial saliva.

### 3.2. XRD

Phase analyses carried out using the X-ray diffraction method showed a slight effect from the thermocyclic aging process and the chewing simulation process of the tested ceramic materials on their phase composition. The HighScore software was used to identify the qualitative and quantitative composition of the phases. The exemplary X-ray diffraction pattern for tested zirconia material after sinterization and oral 10 years simulated chewing environment is presented in Figure 5.

The monoclinic content in the aged samples was proportional to the aging time; in the case of chewing simulations, the monoclinic (−111) content was higher for the samples that were simulated chewing under a load of 70 N. As can be seen in Figure 6, as a representative of all research variants, the tetragonal peak t1 (transformed) was quantitatively dominant, characterized by a smaller unit cell volume and smaller lattice strains. The content of the stabilizer in the case of the t1 peak and the t2 (untransformed) peak was in the range of 1.5–2 vol.% and 5.7–6.5 vol.%, respectively (in the range of the measuring accuracy of ±1 vol%). A clear cubic peak (111) was recorded for the case of a sample after 10,000 cycles of thermo-aging.

The monoclinic contents in crown surface V_m_ are shown in Figure 7. The results represent the transformed fraction depending on the research variant.

The t1 phase appearing in the case of the B3_in_chs_70N had the highest c/a ratio = tetragonality ~1.0159. The t2 phase had a significantly lower tetragonality, ~1.007. Equally high values were recorded for the maximum simulation time (B4_in_chsx2_50N). The t1 phase revealing for that research test variant had the c/a ratio = tetragonality ~1.0158 and the t2 phase ~1.0065. According to data provided by [6], a tetragonality of 1.015–1.016 (c = 517.7 pm) corresponds to a yttria content of 2.5 mol% Y_2_O_3,_ which is the composition at the t/t+c phase boundary.

An analysis of the width of the t1 and t2 peaks has shown that the t1 peaks were narrower and that the domains of the t1 phase were, in that way, initially larger than the domains of the formed t2 phase.

### 3.3. Thermal Etching

SEM images of ceramic crown surfaces polished and thermally etched at T_te_ = 1300 °C are shown in Figure 8.

The scanning microscopy observations revealed a fine-grained, homogeneous structure. Zirconium oxide tetragonal crystals form fine grains of 0.2–0.8 microns, regardless of research variants. The material has a low porosity, which in the tetragonal form also means that it is a non-absorbent material. The high density (approx. 6 g/cm^3^ after sintering) and the lack of porosity of the surface translates into a reduced risk of cracking or fracture.

## 4. Discussion

One of the fundamental manufacturing technologies of today is computer-aided design/computer-aided manufacturing (CAD/CAM), resulting from the progress and rapid development in both dental material production and computer engineering. The production of all-ceramic restorations based on CAD/CAM technology is the most advanced technological process used in dentistry.

The zirconium oxide ceramic substructure provides the highest mechanical strength among all dental ceramics and is used for the reconstruction of defects also in the lateral sections of the dental arch. Thus, it allows us to achieve high requirements of functionality and aesthetics. However, leaving the framework material exposed in the oral cavity may accelerate its low-temperature degradation and increase plaque build-up [44]. There are also doubts as to whether it is advisable, at the current level of knowledge, to use all-ceramic restorations in patients with occlusal overloads caused by parafunction. As is known, the physiological occlusal forces generate pressure in the range of 50–250 N, while with bruxism, the values of these forces increase to 1000 N [7]. That’s why the possibility of an unpleasant incident’s occurrence (fracture, swallowing) has led the scientific and prosthetic community to deeply study the behavior of zirconia, especially by paying attention to aging in vitro and in vivo. Current knowledge shows the strong variability of zirconia in vitro and in vivo degradation as a consequence of the strong influence of processing on the aging process [35,36,37,38,48].

The need for information about the fatigue characteristics of dental materials before they are applied in clinical practice has led to the development of a range of devices designed to simulate chewing. This study is important because fatigue is the weakening of a material caused by repeatedly applied loads. It could be the progressive and localized structural damage (changes in microstructure, microscopic cracks) that occurs when a material is subjected to cycling loading and unloading. These simulators are used before final load testing to provide information about a material’s behavior during long-term use.

Unfortunately, so far, no standards have been developed that would clearly define the conditions for conducting tests determining both wear and aging of the material, as is the case when measuring fracture resistance.

Since the property of zirconium oxide is known, which under the influence of generated mechanical stresses may be accompanied by transformation (reverse transition from tetragonal cell to monoclinic), as well as spontaneous transformation into the monoclinic phase in humid conditions [13], the yttrium-stabilized zirconia material was tested in this work (5Y-TZP) for phase composition changes (XRD method) and microstructure changes (SEM observations) under the influence of generated loads and temperature changes, simulating the typical functionality of tooth crowns during food chewing. Their resistance to cracking was assessed, which is an indicator of a safe and aesthetic role for as long as possible.

Crowns have been tested for their thermal shock resistance because they sometimes have to face great temperature differences between their inner and outer surfaces (e.g., eating ice cream or drinking hot coffee). In addition, they also have to bear bite forces (loads) simultaneously. In order to simulate these phenomena, a thermocycler with the capability to perform thermocycling between a cold and hot tank was used. This was useful to age the samples artificially, as the rapid changes in temperature which could initiate small cracks in the material. Moreover,
Phase tests carried out using the X-ray diffraction method showed a slight effect on the aging process, as shown in Figure 5 and Figure 6. The content of the monoclinic phase also changed to a small extent depending on the load variant. The maximum share (8.19%) was recorded for the crown subjected to the greatest load of 70 N (sample in the initial state, not subjected to aging), as shown in Figure 7;Since the fluctuations of the Vm share were in the range of 2.46–8.19% (with the increase in the number of cycles), it was considered that an increase in stress results in greater but still moderate changes in phase composition of the stressed sample surface, after the accelerated aging processes. That’s why the simulation of conditions reflecting the aging and physiologically chewing process retained crowns with their proper structural integrity, which is a determinant of functional, long-term, and safe use in the oral cavity environment. The crowns stayed complete, as demanded by the ISO standard [23]. That phase changes did not affect the functionality of the prosthetic device;Visual assessment of the grain size and to identify potential cracks in the near-surface zone of the crowns, resulting from the appearance of possible structural instability during testing, did not reveal significant differences depending on the research variables, as shown in Figure 8. All crowns showed only characteristic wear resulting from friction between the surfaces (contact with the antagonist is shown in Table 2).

Since the conducted research was preliminary, some thoughts regarding future research come to mind. First, the use of more reliable forms of the antagonist (zirconia or dentine). Second, the additional use of the OIM (Orientation Imaging Microscopy) method enables the characterization of polycrystalline materials in terms of crystallographic relationships (e.g., determination and distribution of phases, maps of orientation, and grain size).

## 5. Conclusions

Within the limitation of this study, the following conclusions can be drawn:The simulation of the conditions reflecting the chewing process showed that the crowns retained their correct structure, which is an indicator of the proper transfer of loads during chewing, even after accelerated aging;Microscopic observations did not reveal the presence of microcracks in the structure of the examined crowns, which may confirm their safe and long-term use in the oral cavity environment;Progressive transformation of ZrO_2_ grains from the tetragonal phase to the monoclinic phase in the presence of artificial saliva solution (transformation t–m) depended on the number of aging cycles and the value of the force loading the crown. The share of Vm increased with the increase in the number of cycles in the range of 2.46–8.19%.

## Figures and Tables

**Figure 1 materials-16-01171-f001:**
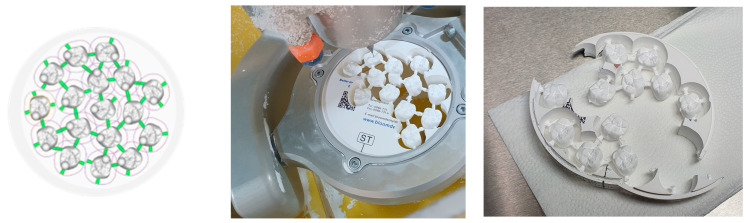
CAD/CAM preparation of zirconia crowns.

**Figure 2 materials-16-01171-f002:**
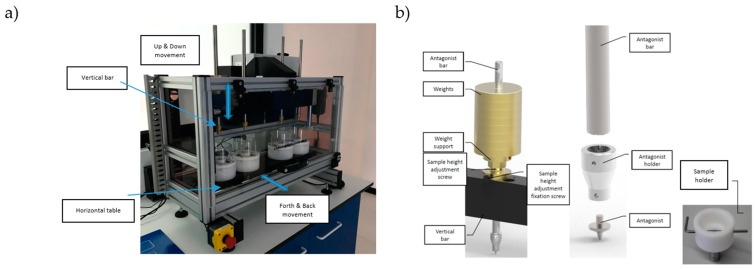
Setup of wear simulations: (**a**) chewing simulator with moving directions, (**b**) antagonist holding system (overview left and detailed view right).

**Figure 3 materials-16-01171-f003:**
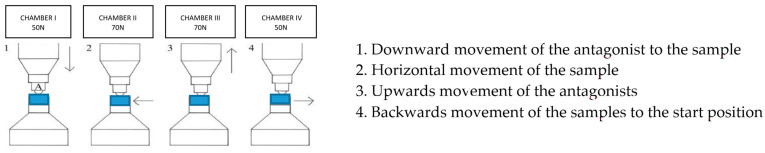
The algorithm of linear movement of a chewing simulator and the distribution of the load in the test chambers: An Antagonist, Crown 

.

**Figure 4 materials-16-01171-f004:**
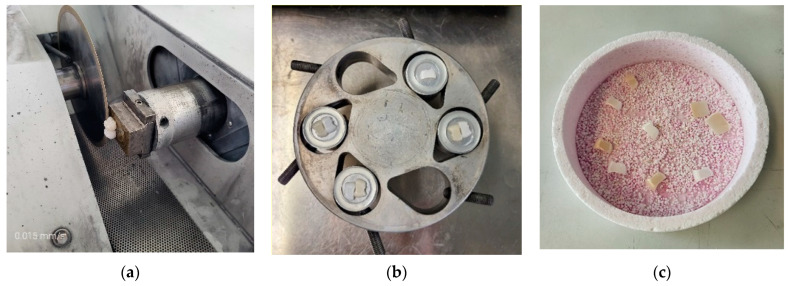
Steps of preparing crowns for thermal etching: (**a**) cutting samples, (**b**) parts of crowns fixed with temporary glue for grinding and polishing, (**c**) finished samples.

**Figure 5 materials-16-01171-f005:**
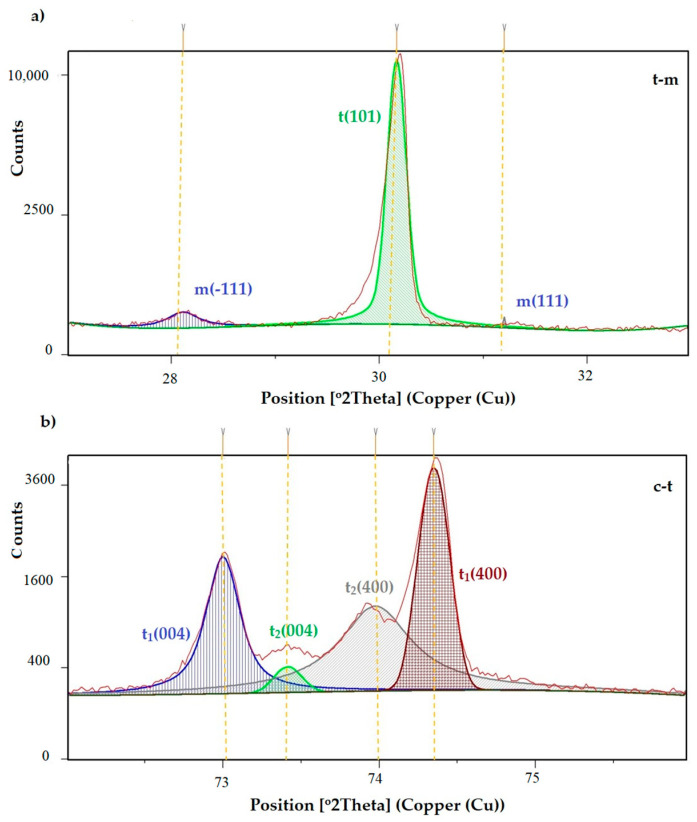
(**a**) t-m, (**b**) c-t transformation of the crown B4_in_chsx2_50N.

**Figure 6 materials-16-01171-f006:**
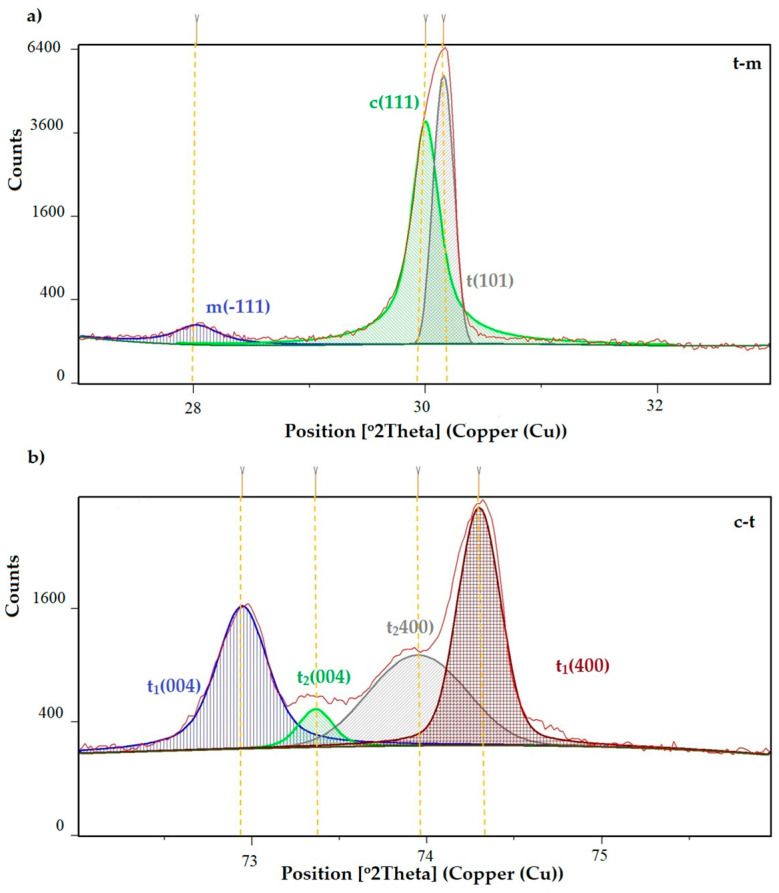
(**a**) t-m, (**b**) c-t transformation of the crown B10_10,000.

**Figure 7 materials-16-01171-f007:**
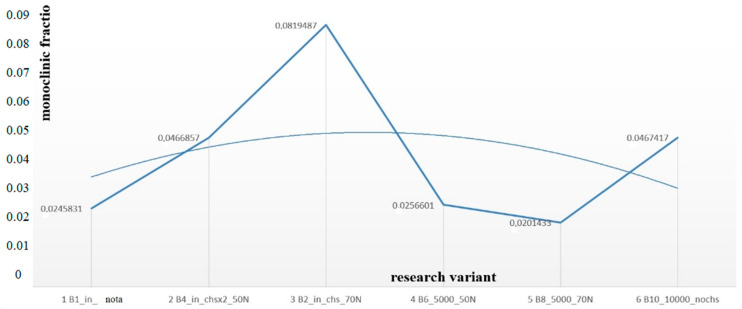
Monoclinic contents of Vm in zirconia material depended on research variables.

**Figure 8 materials-16-01171-f008:**
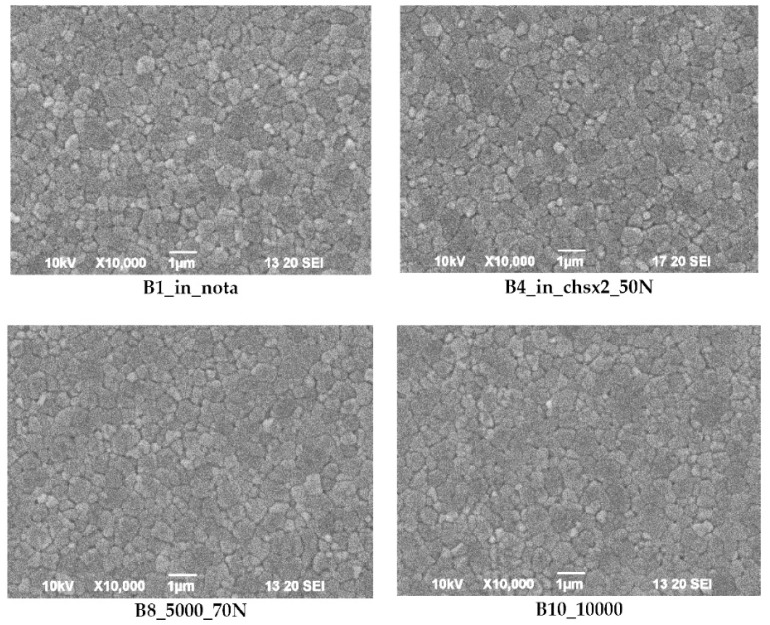
SEM images of thermally etched crown microstructures.

**Table 1 materials-16-01171-t001:** Research variants and the legend of the designations of the samples subjected to the simulation.

Sample Designation	Research Variant			Chewing Simulation		
	Sinterization	Thermal Aging Cycles Number	Chewing Simulation	Chamber Number	Load, N	Cycles Number
B1_in_nota	+	-	-	-	-	-
B3_in_chs_70N	+	-	+	III	70	1,200,000
B4_in_chsx2_50N	+	-	+	IV	50	1,200,000 + 1,200,000
B6_5000_50N	+	5000	+	I	50	1,200,000
B8_5000_70N	+	5000	+	III	70	1,200,000
B10_10,000	+	10,000	-	-	-	-

**Table 2 materials-16-01171-t002:** Wear patterns of the zirconia crowns caused by contact with steel antagonists.

Sample (Chamber) Number and Research Variant	Chewing Simulation Load [N]	Stereo Zoom Microscopy		SEM
1 (I) initial state (sinterization) B1_in_nota	50	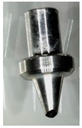 post-chew antagonist	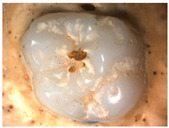 7.5×	* 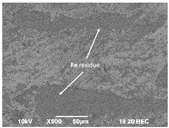 * tracks of steel antagonist
B3 (III) initial state (sinterization) B3_in_chs_70N	70	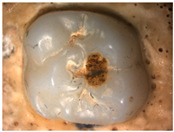 7.5×	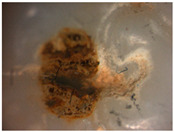 32.5×	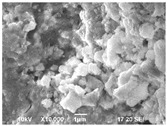 powder layer remained after CAM process
B4 (IV) initial state (sinterization) 1,200,000 cycles ×2 B4_in_chsx2_50N	50	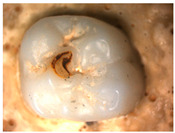 7.5×	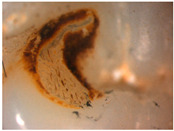 32.5×	* 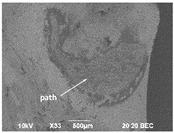 * path of occlusal contact
B6 (I) 5000 cycles of thermocyclic aging B6_5000_50N	50	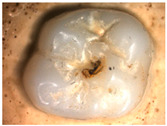 7.5×	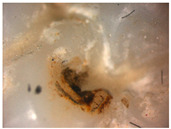 20×	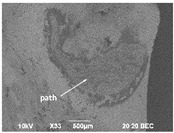 visible cracks in the powder layer
B8 (III) 5000 cycles of thermocyclic aging B8_5000_70N	70	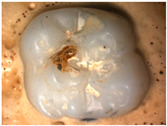 7.5×	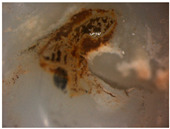 32.5×	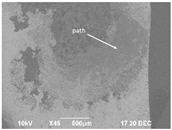 path of occlusal contact

## Data Availability

The data presented in this study are available on request from the corresponding author.

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
