# Peer review of "Effect of Simulated Mastication on Structural Stability of Prosthetic Zirconia Material after Thermocycling Aging"

_materials, 2023, doi:10.3390/ma16031171_

Round 1
Reviewer 1 Report
The aim of the paper is to evaluate the impact of the cyclic temperature change and the chewing simulation process on the stability of the Yttrium-stabilized zirconia material structure: 5Y-TZP. The topic is up-to-date and corresponds to the journal’s area. The adequate research methods are used. All figures and references are cited in the text and there are not missing references in the list. However, there are a lot of things that should be edited.
1. The structure of the manuscript is not well designed: Discussion and Conclusion chapters are missing.
2. The abstract is not informative enough. There is no aim of the paper, the methods are not clearly stated and the information in the Results does not correspond with the results obtained.
3. In the introduction
a. The aim is missing;
b. Page 2, row 56 check the formula of Y2O3?
c. P.2, r.95 and 96 – what are the units of KIC ?
d. P.3, r.102 – check the equation (1): is it K10 or KIO ?
4. In Materials and Methods:
a. 2.1. samples preparing in CAD/CAM – no need of process explanation (row 136-row 143], it is well known. The authors should give information about the material used, the number, shape and sizes of the samples, the machine and software used for manufacturing of specimens.
b. It will be better if the testing conditions, designation and number of samples are given in table.
c. The sintering regime and the equipment also should be added;
d. Fig. 2 and Fig. 3 – the figure captions are not clear.
e. P.4, r.158 – “… steel antagonist..” Did you use quenched steel? Because the common steel is too soft compared to the sintered zirconia.
5. In Results
a. The pictures in Figures 5, 6 and 7 are not clear enough and the captions can not be read.
b. The SEM pictures in Fig. 8 are very dark and not clear. The clear microstructures at higher magnifications should be presented.
Author Response
"Please see the attachment."

Reviewer 2 Report
This paper reports the aging and chewing simulation and its effects on prosthetic zirconia which is an interesting piece of work. This paper is suggested for publication after the authors have incorporated these changes.
1) In the Introduction, although you have explained some properties related to yttria-stabilized zirconia. But there is a lack of comparison with other available materials such as chromium-cobalt-molybdenum and other ceramic composites. etc. for prosthetics especially.
2) The SEM images such as “tracks of steel antagonist” and “path of occlusal contact” needs some contrast and scale bars need to be enlarged in order to get visibility.
3) Images of light microscopy need to add the scale bars.
4) There is a lack of explanation for different microstructures obtained after thermal etching. The authors should have related previously researched structures with the obtained ones.
Author Response
"Please see the attachment."

Reviewer 3 Report
Comments:
1. It is recommended to revise the abstract which needed to be clearer methods, reveal the significant finding in result and precise conclusion.
2. Line 53:” Mechanical stress during mastication and contact with saliva may cause a cooperative disruption of the surface structure of zirconia crowns.” Add Reference(s).
3. Line 57~58, too many parentheses and space. Please correct it.
4. Line 67, [815], Please correct it.
5. Line 91, et. al., please correct it.
6. Line 122, denture?
7. The “Introduction” needs to be organized better. The purpose of this study needs to be clarified.
8. Materials and methods: it is recommended to show the study workflow.
9. Line 175 …”.after aging (for 5000 and for 10000 thermocycles)”, please add reference.
10. Materials and methods What is the material in this experiment?
11. Figures 1 and 2 did not cite in the text, please check them.
12. In line 183, what is the detailed information on light microscopy used in the study?
13. Summary: It is recommended to add a discussion paragraph which could reveal the meaning of the results.
Author Response
"Please see the attachment."

Reviewer 4 Report
In 2.1Samples preparing - subsection there is no specific details of manufacturing (process parameters). The description is to general and can be placed in the introduction. How many samples were produced and tested?
· - In 2.2. Thermocycling aging – a thermal cyclogram showing also the time of heating-maintain-cooling should be presented.
· - In 2.3. Chewing simulation – it is not clear whether each sample is passing through all four chambers and how many loading cycles are completed in each chamber.
· - The figure 5 cannot be read, the text is too small.
· - The SEM images in the figure 8 are really hard to read and interpret. Some marks in the images (or detail picture) indicating the explanations in the lines 254-258 should be inserted.
· - The stress acting in the dental structure is directly influenced by the angulation of the load and the loafing modulus. Some simulations on this topic indicate how stress is propagating in dental crowns and therefore a discussion in your context of cycling loading can be introduced. Here are some references:
Alberto, L.H.J.; Kalluri, L.; Esquivel-Upshaw, J.F.; Duan, Y. Three-Dimensional Finite Element Analysis of Different Connector Designs for All-Ceramic Implant-Supported Fixed Dental Prostheses. Ceramics 2022, 5, 34-43. https://doi.org/10.3390/ceramics5010004
Chirca, O.; Biclesanu, C.; Florescu, A.; Stoia, D.I.; Pangica, A.M.; Burcea, A.; Vasilescu, M.; Antoniac, I.V. Adhesive-Ceramic Interface Behavior in Dental Restorations. FEM Study and SEM Investigation. Materials 2021, 14, 5048. https://doi.org/10.3390/ma14175048
Kalluri, L.; Seale, B.; Satpathy, M.; Esquivel-Upshaw, J.F.; Duan, Y. Three-Dimensional Finite Element Analysis of the Veneer—Framework Thickness in an All-Ceramic Implant Supported Fixed Partial Denture. Ceramics 2021, 4, 199-207. https://doi.org/10.3390/ceramics4020015
Author Response
"Please see the attachment."

Round 2
Reviewer 3 Report
Article: Effect of Simulated Mastication on Structural Stability of Prosthetic Zirconia Material After Thermocycling Aging
No.: materials-2120594
Comments:
1. Although the abstract has been modified, it should be strengthened AGAIN based on the experimental results and the conclusions in line with the title, so that readers cold have a clearer understanding of the scientific findings of this study.
2. The order of references needs to be rechecked and adjusted.
3. In the discussion from lines 341 to 354, the authors described the advantages of FEA, however, it needed more relevance to this paper. The author should compare the FEA results of the reference with the results of this experiment so the discussion can be more meaningful.
4. The authors did not revise well to our 1st all questions.
Author Response
Please see the newest version of the manuscript.
Thank you.
